# Position: Benchmarks Cannot Establish Deployment Readiness of Clinical AI

Haoran Zhang [* 1]  Hyewon Jeong [* 1]  Olawale Salaudeen [* 1]  Walter Gerych [* 2]  Nigam Shah [3]
Marzyeh Ghassemi [1]

## Abstract

Despite large language models (LLMs) achieving impressive performance on benchmark tasks such as medical question answering, their real-world utility remains limited. We argue that while benchmarks play a valuable role in developing methods and filtering promising models during development, they fundamentally cannot establish deployment readiness. Many models topping benchmark performance have failed to perform seemingly related clinical tasks effectively in practice, while others with modest benchmark performance have demonstrated meaningful clinical benefits. We detail the limitations of benchmark-centric evaluations of deployment readiness. We argue that we should use benchmarks only to identify candidate methods or models, not to justify deployment. We call for increased use of other measurement instruments of deployment readiness, such as prospective studies, and policy changes that align incentives with clinically grounded evaluation.

## 1. Introduction

Artificial intelligence (AI) is often touted as a means of transforming healthcare. However, the path from model prototypes to real patient impact is neither straightforward nor guaranteed (Zhang et al., 2022; Char et al., 2020). One roadblock is the challenge of evaluating whether models have clinical utility in real-world use. Common measurement instruments for health AI systems are *decontextualized general health-AI benchmarks* (which we refer to as benchmarks moving forward): task-dataset-score triples, often paired with leaderboards, that evaluate general AI capabilities in a reproducible manner (Jin et al., 2021; 2019; He

et al., 2020; Pal et al., 2022; Zhang et al., 2023; Gupta & Demner-Fushman, 2022; Johnson et al., 2016; 2024a; Pollard et al., 2018; Johnson et al., 2024b). One example is MedQA (Jin et al., 2021), a large-scale open-domain question-answering dataset composed of medical exam-style questions. Benchmarks like MedQA are valuable because they make model comparisons reproducible and allow specific capabilities to be tested under controlled conditions. The problem we highlight is that these measurements are often treated as evidence for broader deployment claims than they can support.

Performance on exam-style benchmarks need not predict effective use in clinical workflows. A model scoring high on such benchmarks may not improve clinical decisions (Goh et al., 2024), integrate safely into clinical workflows (Omar et al., 2025), generalize across patient populations and care settings (Yang et al., 2024a; Omar et al., 2024; Rahman et al., 2024; Pfohl et al., 2024), or produce net benefit for patients (Lam et al., 2022; Han et al., 2024). The mismatch between what general health-AI benchmarks measure and the claims they are used to justify motivates a simple but urgent question: **what role should benchmarks play in deciding whether we place AI tools in front of clinicians and patients?**

We define *clinical utility* broadly as the extent to which an AI system, when deployed in practice, improves clinical outcomes, safety, efficiency, or decision-making. This is not an exhaustive definition (see Figure 1b). Rather, it provides a minimal target for our argument: if benchmarks struggle to support even this broad notion of utility, the concerns we raise become only more pronounced under richer, multi-stakeholder formulations of utility and setting-specific objectives. Clinical utility is also a multidimensional and context-sensitive target. This does not make it unsuitable for rigorous evaluation. Many constructs used in high-stakes sciences, including psychology, education, and social science, are similarly complex, but can still be evaluated when measurements are explicitly tied to the claims they are meant to support. The problem is not that clinical utility is difficult to define once and for all, but that benchmark performance is often treated as if it resolves this difficulty.

This problem is not unique to large language models

---

*Equal contribution, author order determined by an exhilarating 1D random walk with author-fixed random seeds.    [1]MIT [2]Worcester Polytechnic Institute [3]Stanford University. Correspondence to: Olawale Salaudeen <olawale@mit.edu>, Haoran Zhang <haoranz@mit.edu>.

*Proceedings of the 43rd International Conference on Machine Learning*, Seoul, South Korea. PMLR 306, 2026. Copyright 2026 by the author(s).

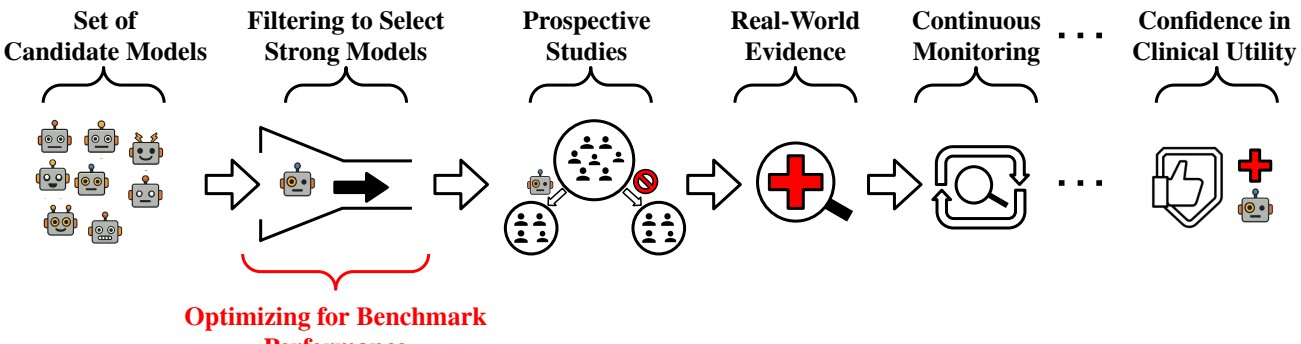

*(a)* **A proposed pathway for clinical AI development**, where benchmarks are used only as an initial filter. To establish true utility, models must undergo rigorous prospective validation in real-world use, coupled with continuous monitoring of performance and outcomes to collect the evidence establishing utility.

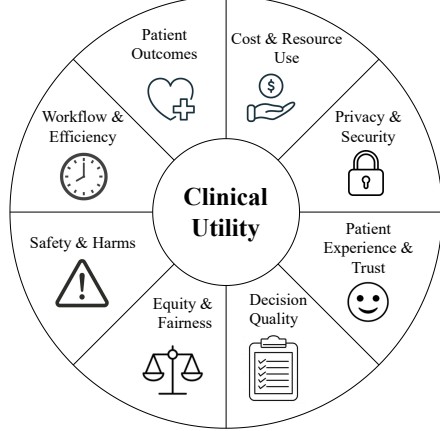

*(b)* **Clinical utility may be defined along many dimensions**; the factors shown here are illustrative examples rather than a comprehensive taxonomy.

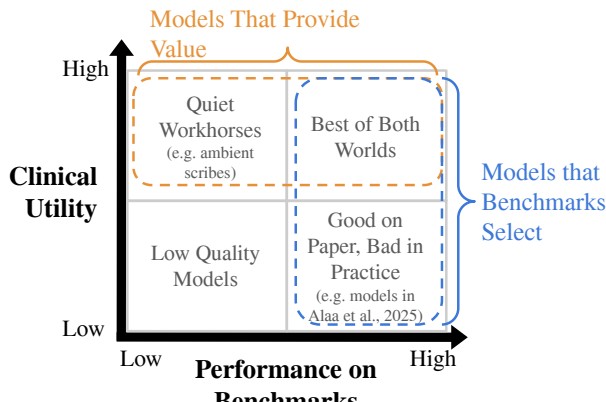

*(c)* **Current landscape showing the *mismatch* between benchmark performance and clinical utility.** Many models selected for deployment based on strong benchmark results (blue box) fail in real-world settings (Good on Paper, Bad in Practice, such as those evaluated in Alaa et al. (2025)), while some clinically valuable models are overlooked due to modest benchmark scores (Quiet Workhorses, orange box).

*Figure 1.* Beyond Benchmarking: Toward Clinically Useful AI Models.

(LLMs). Several high-profile cases from earlier generations of clinical AI illustrate that strong benchmark performance can fail to translate into real-world impact (Figure 1). IBM Watson for Oncology, for instance, achieved strong performance on benchmarks of expert treatment recommendations but still generated unsafe or irrelevant suggestions when integrated into clinical workflows (Ross & Swetlitz, 2018; Dolfing, 2024; Zhou et al., 2019). Similarly, Epic's proprietary sepsis model was adopted in hundreds of hospitals based on promising benchmark performance but missed two-thirds of actual sepsis cases when externally validated (Wong et al., 2021; Kamran et al., 2024). These examples show that benchmark success alone does not establish that a system will function safely or usefully once embedded in clinical practice.

In this work, we focus on LLMs, a dominant paradigm

in contemporary health AI. LLMs such as GPT and Med-PaLM have achieved near-human or superhuman performance on medical question-answering benchmarks (Nori et al., 2023; Singhal et al., 2023). Yet these models continue to hallucinate, misstate clinical facts, and fail to support safe decision-making in deployment-relevant settings (Chen et al., 2024b; Omar et al., 2025). As consumer-facing and health-system-facing LLM tools enter clinical environments, this gap between benchmark performance and clinical utility becomes increasingly consequential. Consumer-facing and health-system-facing LLM tools are increasingly piloted in clinical environments (OpenAI, 2026); however, one independent study found that it undertriaged 52% of true emergency scenarios and overtriaged 65% of nonurgent cases, illustrating that current frontier systems can still fail at the clinical decision points where deployment safety matters most (Ramaswamy et al., 2026).

Beyond triage, LLMs can also fail in ways that are poorly captured by static medical benchmarks. They have been shown to propagate gender and racial bias in clinical decision-making (Rickman, 2025; Benkirane et al., 2025). In an evaluation of 21 reasoning LLM variants across 8 frontier models, exposure to misleading clinician rationales caused significant diagnostic degradation in 14 models, despite strong standalone benchmark performance (Lopez et al., 2026). Users of LLMs also often provide incomplete information, misinterpret model outputs, or fail to act on correct advice (Bean et al., 2026). Together, these failures suggest that the central question is not whether LLMs can perform well on health-AI benchmarks, but whether such benchmarks provide valid evidence for the clinical claims being made about them.

These examples represent two broad and common failure modes: (i) the distributions that models are benchmarked on differ from the distributions they encounter in practice, and (ii) models are benchmarked independently of the workflows in which they must operate in practice.

While recent stress tests show that state-of-the-art LLMs can top general health benchmark leaderboards while remaining severely limited in their desired clinical utility (Gu et al., 2025; Bedi et al., 2025b), the reverse is also true. Models that do not chart on general benchmark leaderboards can still have substantial clinical utility when they are designed for a specific workflow and evaluated against the right endpoint. Ambient AI scribe systems are one example. They are not primarily useful because they answer medical exam questions, but because they may improve documentation efficiency and reduce clinician burden in real clinical settings (Feldheim, 2025; Pearlman et al., 2025; Stults et al., 2025; Lukac et al., 2025; Afshar et al., 2025). Evaluations of scribes also surface important limitations, including documentation errors, hallucinated or omitted information, variable note quality, and unresolved accountability risks (Anderson et al., 2025; Palm et al., 2025; Topaz et al., 2025). Scribes are therefore a useful example of why clinical utility must be assessed directly rather than inferred from general benchmark performance.

These examples underscore a core insight: **high decontextualized general health benchmark performance is neither necessary nor sufficient for clinical utility.** Overreliance on benchmark scores risks misleading researchers, clinicians, and policymakers about the utility of models in practice. This remains true even as benchmarks become more interactive, multimodal, and broadly grounded in clinical tasks. Such benchmarks can better screen candidate models, but they still evaluate model behavior outside the organizational, temporal, and human decision-making contexts that determine real-world clinical utility.

We therefore propose a narrower role for benchmarks: **benchmarks should be used to select promising candidate methods and models under general, well-defined technical criteria. Claims about clinical utility should be evaluated through real-world prospective studies of models in deployment.** Rather than centering general-purpose foundation models and their benchmarks, research resources would be better spent on evaluations that measure utility in the settings where models are intended to be used, instead of proliferating redundant general-purpose benchmarks that conflate technical performance with clinical value.

## 2. The Mismatch Between Benchmark Tasks and Clinical Tasks

In this section, we highlight sources of misleading conclusions from benchmark performance.

**Dataset Flaws.** Clinical benchmarking datasets frequently contain various data quality issues, such as insufficient patient information, incorrect or outdated labels, and ambiguous questions or answers. For example, one study found that 4% of the questions in MedQA contain insufficient information to answer, and an additional 3% have label errors (Griot et al., 2025). The authors of PubMedQA estimate that up to 1% of the answers may be mislabeled (Jin et al., 2019), which is larger than the performance difference between top models on this dataset. In medical imaging, disease labels from chest X-rays are often derived by automated labelers operating on radiology reports, which have been found to have high error rates (Smit et al., 2020). Reaching maximum performance on these benchmarks despite data errors can thus reward exploitation of spurious correlations or memorization, behaviors that are disconnected from clinical utility.

**Memorization** LLMs may have seen benchmark questions or answers during pre-training, artificially inflating their performance. This test data contamination has been well-documented in general NLP benchmarks such as MMLU (Deng et al., 2023; Yang et al., 2023). Even when test questions are not explicitly present in the training set, LLMs may overfit to benchmark formats or metadata cues, further boosting scores for the wrong reasons (Deng et al., 2023). In the clinical domain, models like GPT-4 have succeeded on exam-style benchmarks such as the USMLE (Kung et al., 2023), but fail simply when the correct answer is replaced with "None of the other answers" in the original questions. This memorization may also be responsible for failures in open-ended clinical tasks, such as summarizing longitudinal charts (Liu et al., 2024). Clearly, strong benchmark performance may reflect memorization more than generalizable reasoning ability. Additionally, we must consider Goodhart's Law ("when a measure becomes a target, it ceases to be a good measure") (Zhang et al.,

2024a).

**Dynamic Nature of Clinical Decision Making is Missed.**
Real-world clinical decision-making is a dynamic, interactive process involving hypothesis generation, data gathering, and iterative refinement of diagnoses. Clinicians typically start with a patient's complaints and then iteratively inquire about additional symptoms, review history, conduct exams, or order tests. Simultaneously, clinicians maintain a differential diagnosis, a list of plausible diseases given a patient's condition, that is refined as new information emerges (Guyatt & Rennie, 1993). This multi-step reasoning process acknowledges uncertainty. Clinicians rarely commit to a single diagnosis at the outset; instead, they collect sufficient evidence to rule out other diagnoses before reaching a final diagnosis or treatment plan. Recent work on LLM-assisted differential diagnosis suggests a more clinically aligned role for AI, helping clinicians broaden the diagnostic search space rather than forcing a single answer (McDuff et al., 2025).

Widely used medical QA benchmarks drastically simplify this process (Jin et al., 2019; 2021; Arora et al., 2025; Chen et al., 2024a; Bae et al., 2023; Raji et al., 2025). Designed in the style of standardized exams, these benchmarks test specific facts from predefined answer choices based on static clinical vignettes. Very few are derived from real-world clinical data (Bedi et al., 2024). The model cannot ask clarifying questions, seek additional data, or reason through an evolving care context. Moreover, these tasks are often defined retrospectively and correspond to a single snapshot in the care process, rather than capturing the longitudinal and adaptive reasoning clinicians use at the bedside (Li et al., 2024b). Newer interactive benchmarks have begun to partially address this gap (Li et al., 2024a; Chiu et al., 2026), but they still simplify the active clinical setting: interactions are constrained, patient information is predefined, workflow pressures are absent, and success is measured against retrospective labels rather than downstream clinical outcomes.

This mismatch limits what benchmark performance can tell us about clinical utility. While such benchmarks are useful for identifying promising models, they fall short of evaluating models that could, in principle, support dynamic, iterative clinical decision-making. Strong benchmark performance, therefore, does not necessarily translate into improved clinical utility. Many AI systems that achieved high retrospective accuracy have shown no significant benefit in prospective clinical outcomes compared with standard care (Zhou et al., 2021). In practice, the most useful model is not always the benchmark or theoretical champion, but the one that satisfies real-world operational constraints and earns stakeholder trust (Sandhu et al., 2020) (Figure 1 (c)).

**Modalities of Clinical Information are Missing.** Many clinical AI evaluations still occur in contextually stripped settings, as they operate on narrow, often unimodal representations of a patient's state, such as clinical notes, structured EHR data, or radiology images, and lack access to the full range of information available in clinical environments (Tikhomirov et al., 2024). For example, board-style questions are typically carefully curated textual vignettes that compress a patient's disease course into a static, exam-oriented summary (He et al., 2020; Zhang et al., 2023; Gupta & Demner-Fushman, 2022; Ikezogwo et al., 2023). In contrast, human clinicians use a much richer set of features, not only combining labs, imaging, and notes (Wang et al., 2024), but also incorporating physical examinations, behavioral cues, prior clinical experience, and interactive patient engagement.

This gap creates an information bottleneck, in which the inputs available to AI systems are incomplete and compressed relative to the contextual information available to clinicians. Models trained on limited modalities may therefore rely on shortcuts such as artifacts, documentation patterns, or site-specific cues that inflate benchmark performance but do not generalize in practice (Richens et al., 2020; Mahmood et al., 2021; DeGrave et al., 2021; Caruana et al., 2015). Human clinicians are not immune to bias or shortcut reasoning, but they often have additional context, clinical priors, and opportunities for verification that help them discount clinically irrelevant signals. For instance, a human clinician is unlikely to treat a site-specific image marker as evidence of disease, whereas a model trained on decontextualized data may learn that association if it improves benchmark accuracy (DeGrave et al., 2021). General benchmarks rarely test whether models can distinguish causal clinical relationships from dataset-specific correlations or verify learned relationships through additional evidence. As a result, missing modalities do not simply reduce the amount of information available to a model; they can change what the model learns, making high benchmark performance compatible with brittle behavior in novel clinical settings.

**Oversimplified Evaluation Metrics.** Existing medical LLM benchmarks are almost entirely focused on performance quantified by a single numeric metric, often aggregating over many important individual components (Salaudeen et al., 2025b). For example, MedQA (Jin et al., 2021) and PubMedQA (Jin et al., 2019) use multiple-choice accuracy; other benchmarks (Abacha et al., 2023; Zhou et al., 2025; Yim et al., 2023; Zhang et al., 2024b) use BERTScore (Zhang et al., 2019), despite known limitations (Hanna & Bojar, 2021); and fairness benchmarks look at output sensitivity to demographic changes (Pfohl et al., 2024; Zhang et al., 2024c). However, almost no benchmarks consider multiple dimensions at once. In medicine,

optimizing one metric can degrade another—performance might trade off with fairness, safety, clarity, or explainability (Kleinberg et al., 2016; Menon & Williamson, 2018; Chouldechova, 2017).

By focusing on overall accuracy, we risk overlooking when models give misleading explanations (Sim & Chen, 2024) or perform poorly on subpopulations (Poulain et al., 2024; Salaudeen et al., 2025b). GPT-4, for instance, achieved high scores on licensing exams (Nori et al., 2023), but exhibited accuracy drops when cognitive biases were introduced (Schmidgall et al., 2024), and reinforced racial and gender stereotypes (Zack et al., 2024). A comprehensive evaluation should be multi-objective (e.g., utility, safety, fairness, timeliness), summarized via composite indices or Pareto front analyses, rather than a single metric. Both aggregated and disaggregated evaluations are needed (Pfohl et al., 2025).

The contrast with how human physicians are evaluated is stark. Doctors are assessed on communication, ethics, decision-making, and patient outcomes, not just written exams (Epstein & Hundert, 2002). A clinician might ace their boards but fail in practice due to poor bedside manner or biased reasoning. Similarly, LLMs should be judged by holistic standards. Simply improving benchmarks, while helpful in narrow cases, delays evaluations that are truly informative for clinical utility.

**Local Utility vs. Generalizability**  Local utility refers to whether a model improves decisions, workflows, or outcomes in a specific clinical setting. Generalizability refers to whether its performance holds across settings, populations, or time. They need not go hand in hand. High generalizability does not ensure, and may not be necessary for, clinical usefulness (Stanford Institute for Human-Centered Artificial Intelligence, 2024). Benchmarks built around broad latent traits such as "medical reasoning" or "clinical understanding" are often designed to be generalizable by definition. They are intended to transcend context and apply across domains, settings, and populations. A model that performs well on these tasks appears to be broadly capable and therefore deployable across many clinical situations (Nori et al., 2023; Achiam et al., 2023; Singhal et al., 2023).

This view overstates what benchmarks can support. The generalizability of benchmark performance is not the same as the generalizability of clinical impact. A model may generalize across synthetic question-answering tasks, related datasets, or benchmark domains, but still fail to deliver meaningful benefit in practice (Chen et al., 2024b; Wornow et al., 2023). Similarly, a model may produce answers that align with annotator expectations yet fail to support the decisions clinicians need to make in high-stakes care (Chen et al., 2024b). The key question is not only whether performance transfers across datasets, but whether the model improves decisions, workflows, or outcomes in the settings where it is used.

The reverse is also true. Models with limited benchmark generalizability may still provide local utility. A triage model tailored to one hospital's resource constraints might meaningfully reduce emergency department overcrowding, even if it performs poorly at another hospital (Sendak et al., 2020; Ghassemi et al., 2019). This illustrates that utility is a property of context-specific interventions, not only context-invariant predictors. Generalizability concerns maintaining performance across applicable settings. Local utility concerns whether a model helps in a particular setting, for a particular task, under particular constraints. It is therefore not the case that broad generalizability should always be a necessary condition for evaluating models, given that substantial value may come from contextual models designed for specific use cases.

Conflating local utility with generalizability risks producing systems that are broadly performant but clinically irrelevant. Benchmarks that support claims of generalization should not be treated as evidence of clinical utility unless they also demonstrate that these generalizations translate into real-world benefits across locations, patient populations, and time periods. If the goal is generalizable clinical utility, then we need generalizable evaluations: multi-site studies, diverse deployment trials, and context-aware impact assessments. In many cases, however, the better goal may be context-aware adaptation rather than universal generalization. Clinical AI should not only ask whether a model works everywhere, but whether it can be made useful where it is actually deployed.

Taken together, these failure modes show why benchmark performance can be misleading evidence for clinical utility. Benchmarks often evaluate models on static inputs, limited modalities, simplified metrics, and claims of generality that abstract away from the clinical settings in which models are used. These limitations do not make benchmarks useless. They make their evidentiary role narrower. Benchmarks can help identify promising models, reveal specific capabilities, and screen for obvious failures under controlled conditions. However, they cannot establish that a model will improve decisions, fit into workflows, reduce harm, or produce net benefit for patients. Those claims require evaluations that are tied to the intended use case, measured against clinically meaningful endpoints, and conducted in the settings where the model is meant to operate.

## 3. Limitations of Benchmarks are Inherent

Even if every shortcoming in Section 2 were addressed, general benchmarks would still face intrinsic limits. The

problem is not only that current benchmarks are incomplete. The deeper issue is that benchmarks are fundamentally limited tools for establishing use-case-specific clinical utility. Clinical utility depends on the intended use, deployment context, human users, institutional constraints, and the model's downstream effects. These factors cannot be fully resolved by making benchmarks larger, more interactive, more multimodal, or more clinically grounded.

**Unrealizable Tasks.** Many benchmark tasks are unrealizable as clinical tasks because real-world clinical problems often do not have a single unambiguously correct answer. MedQA, for example, asks models to synthesize exam-style vignettes and select the best answer from predefined options. This is a valid educational testing format, but it is distinct from diagnosing or treating a specific patient. The goal is to recover the intended answer under a simplified representation of clinical knowledge, not to make a decision under uncertainty for an individual patient. Discharge summary generation benchmarks similarly assume that the model has access to complete structured and unstructured data and must produce an error-free summary (Hossain et al., 2023). In practice, summaries are produced by clinicians under time constraints, with incomplete information, local documentation norms, and no single objectively correct note.

Requiring benchmark success on such tasks as a prerequisite for clinical validation enforces a backward logic. We are asking models to pass tests that no deployed clinical system would be judged by in isolation. This imposes artificial barriers to entry for models that are well-suited to specific, localized clinical interventions. A system that improves triage accuracy, reduces false alarms, or nudges clinicians toward better prescribing behavior may fail on general QA or summarization tasks and still deliver substantial benefit (Sendak et al., 2020; Stanford Institute for Human-Centered Artificial Intelligence, 2024; Adams et al., 2022).

The problem is not simply that these tasks are difficult. It is that success on them may have little to do with the clinical interventions we actually want models to support.

**Validity.** Previous work has mapped different facets of validity (Jacobs & Wallach, 2021; Wallach et al., 2025; Raji et al., 2021) to different types of inference (Salaudeen et al., 2025a; Lissitz & Samuelsen, 2007; Salaudeen et al., 2025c). Construct validity concerns interpretations about underlying theoretical concepts. In the benchmark setting, this often corresponds to claims that a score measures a latent capability such as "medical reasoning" or "clinical understanding." However, such claims require a level of theoretical understanding and interpretability that AI systems currently lack (Salaudeen et al., 2025a). They are also not necessary for assessing utility. Criterion validity, defined as correlation

with a downstream or validated criterion, provides a more direct basis for asking whether a system enables appropriate and effective decisions in real-world contexts.

The problem becomes clear when benchmarks are used to support claims about clinical utility. Asking whether a model has medical reasoning capabilities based on benchmark scores is a question about a construct. Asking whether a model improves health outcomes is a different question. General-purpose benchmarks often blur this distinction by relying on a cascade of assumptions: performance on task A, such as MedQA, reflects capability B, such as clinical reasoning, which is necessary for behavior C, such as high-quality decision support, which then produces outcome D, such as improved health. Each link in that chain is speculative and nearly always untested (Alaa et al., 2025; Salaudeen et al., 2025a). For clinical utility, the more direct question is not whether the benchmark captures the right latent construct, but whether improvements in the evaluation predict improvements in decisions, workflows, or outcomes in the setting where the model is used.

**Misaligned Incentives.** Benchmark-centrism creates misaligned incentives for both researchers and institutions. Benchmarks reward high scores on narrow, static tasks often removed from actual clinical decision-making. This encourages overfitting to benchmark artifacts (Zech et al., 2018; Oakden-Rayner et al., 2020), the design of artificial tasks whose performance is easy to measure but hard to interpret (Wornow et al., 2023), and the use of models that perform well in controlled conditions but fail when deployed (Wong et al., 2021; Daneshjou et al., 2021). These incentives also privilege novelty and complexity over robustness and usability (Ghassemi et al., 2019). Clinically impactful but less glamorous contributions, such as integrating a model into a workflow or tuning thresholds to reduce alarm fatigue, remain undervalued because they are difficult to capture in benchmark metrics.

Current benchmark norms also reward generalized, context-free solutions. This creates a paradox: systems that could deliver genuine clinical benefit in constrained or localized settings are overlooked as insufficiently generalizable (Ghassemi et al., 2020), while researchers face pressure to develop "generalist" models that perform well on tasks such as MedQA or MMLU (Hendrycks et al., 2020; Singhal et al., 2023), even when those tasks may fail to translate into effective real-world systems.

These incentives make shortcut learning predictable. Cancer classification challenges have rewarded vision models that used surgical skin markings or image borders instead of underlying pathology (Daneshjou et al., 2021); chest X-ray classifiers have overfit to scanner artifacts and laterality tokens (Oakden-Rayner et al., 2020; Geric et al., 2023); and

a widely adopted commercial sepsis model failed to generalize outside its development setting (Wong et al., 2021). These are not edge failures. They are predictable outcomes of optimizing for what is easy to measure rather than what is clinically meaningful. For many state-of-the-art benchmarks, we still lack strong evidence that models are not exploiting similar shortcuts.

**Opportunity Cost.** Benchmark performance has become a primary currency for evaluating clinical AI. Benchmarks serve a useful role in model comparison, but over-reliance on them as proxies for real-world effectiveness distorts research priorities, incentives, and innovation pathways.

The most immediate cost is opportunity cost. Benchmark-centric research absorbs substantial intellectual and financial resources, often at the expense of work that more directly supports clinical translation: implementation studies, human-centered design, prospective evaluation, and post-deployment monitoring (Sendak et al., 2020; Ghassemi et al., 2020). These activities are harder to publish, more expensive to run, and less rewarded under current norms, despite being essential for real-world impact (Wiens et al., 2019; Montani & Striani, 2019). A research ecosystem focused on outperforming static benchmarks, therefore, delays the kinds of progress that matter most for patients.

## 4. Narrowing the Gap

*All benchmarks are 'insufficient'; some benchmarks are useful.* Benchmarks will always fall short of fully capturing clinical utility, yet they remain a valuable tool when used appropriately. The problem is not that we have benchmarks; it is that we have allowed them to substitute for more direct forms of evaluation. Moving forward requires a recalibration of how we design, interpret, and use benchmarks, both within the research community and in policy and deployment contexts.

**Modern Benchmarks Narrow, But Do Not Close, the Gap.** Recent benchmarks have begun to address some of the limitations of static, exam-style medical QA. Multimodal benchmarks such as EHRXQA (Bae et al., 2023) combine structured EHR data with chest X-ray images, allowing evaluation of cross-modal reasoning rather than text-only factual recall. EHR-based QA benchmarks (Fleming et al., 2024; Bae et al., 2023; Mehandru et al., 2025) introduce noisy, multi-format clinical data and require reasoning across encounters. Interactive benchmarks and simulation frameworks further move beyond one-shot answering: early symptom-checker benchmarks (Wei et al., 2018; Xu et al., 2019; Peng et al., 2018) allowed models to gather information from simulated patients, although they often oversimplified the task by focusing on single-disease identi-

fication rather than differential diagnosis; more recent work converts static medical QA into an interactive setting in which models can ask follow-up questions before answering (Li et al., 2024a). AgentClinic similarly recasts medical QA as a sequential diagnostic task with simulated patient interaction, multimodal data collection, and tool use; notably, solving MedQA cases in this sequential format can reduce diagnostic accuracy to below one-tenth of the original static accuracy (Schmidgall et al., 2026). Decision-oriented suites such as CLIBench (Ma et al., 2024) evaluate clinical actions including diagnoses, procedures, laboratory tests, and prescriptions, while broader leaderboards such as MedHELM (Bedi et al., 2026), HealthBench (Arora et al., 2025), and other recent benchmarks (Liu et al., 2024) evaluate a wider array of clinical tasks and quality dimensions.

These efforts are important and should be encouraged: they make benchmarks better tools for model development, stress testing, and preliminary screening. However, they do not eliminate the need for deployment-specific evaluation. Even interactive, multimodal, or clinician-rubriced benchmarks remain controlled, offline evaluations of model behavior under a predefined task interface. They generally cannot measure whether clinicians appropriately rely on model outputs, whether recommendations arrive at the right time in a workflow, whether the system reduces workload or introduces new burdens, how errors propagate through downstream decisions, or how performance shifts after deployment as patient populations, documentation practices, and user behavior change. Modern benchmarks can narrow the gap between benchmark tasks and clinical tasks, but claims about deployment readiness still require context-specific validation through local retrospective evaluation, prospective workflow studies, and post-deployment monitoring.

**Use benchmarks as triage tools.** Benchmarks can play a useful role in narrowing the space of models worth studying further. When carefully constructed and transparently scoped, they can support model selection or serve as stress tests for particular failure modes. However, claims about clinical impact should not rest on benchmark performance alone. Instead, benchmarks should be treated as an initial filter, useful for selecting candidate models for more rigorous evaluation, but insufficient for justifying deployment decisions. Advancement from benchmarks to prospective study should be based on demonstrated clinical need, plausible impact, and available resources, not only on a fixed leaderboard cutoff.

**Design evaluations around decisions.** Where measurement instruments are used, they should be aligned with clinical use cases. This means moving beyond simple prediction and generation tasks and toward evaluations that reflect actual decisions, such as triage prioritization, diag-

nostic referrals, or treatment selection under uncertainty. Evaluation targets should shift from per-example accuracy to metrics that capture marginal benefit, timing, and downstream effects such as patient outcomes and harms. Importantly, evaluations should be assessed for their own validity: what construct they measure, what criterion they correlate with, and what conclusions they support (Alaa et al., 2025; Salaudeen et al., 2025a). Modern evaluation suites have narrowed this gap by assessing AI systems on a broader range of clinical tasks that influence decision-making, although the end-to-end integration of these tasks remains limited (Bedi et al., 2026).

**Test utility prospectively.** Regardless of how well-designed benchmarks are, they remain hopeful correlational proxies for clinical utility. We join recent perspectives (Azad et al., 2026; Agrawal et al., 2025) in arguing for a prospective, real-world evaluation pathway in which study design is matched to risk, context, and maturity of the tool. Such studies allow us to measure patient-centric metrics (e.g., time to diagnosis, patient morbidity, and mortality) and health system metrics (e.g., clinician stress and cognitive load), which are the ultimate objectives of deploying these models in clinical settings. Such objectives cannot possibly be evaluated by a static benchmark. Finally, prospective studies also surface unanticipated harms, such as automation bias, over-reliance, and privacy issues, which benchmarks also do not evaluate. Randomized controlled trials (RCTs) are one option in this toolbox (Han et al., 2024; Goh et al., 2024; Omar et al., 2024), indispensable for some high-risk decisions or when residual confounding cannot be otherwise addressed, but can often be slow and costly. When randomized evaluation is not immediately feasible, target trial emulation can provide a useful intermediate framework for specifying the clinical intervention, time zero, follow-up window, and outcomes workflow (Hernán & Robins, 2016). Even outside RCTs or target trial emulation, prospective and real-world studies offer a clearer picture of how models function when embedded in practice, capturing effects on clinician workflow, safety, and patient trust (Sendak et al., 2020; Nori et al., 2023; Kim et al., 2024).

Emerging evidence underscores the urgency of moving beyond retrospective proof-of-concept work. Systematic reviews of ML-based studies (Ben-Israel et al., 2020) reported that only 2% were prospective, including Adams et al. (2022); Abràmoff et al. (2018); Scheetz et al. (2021), with most research focused on retrospective analyses. To systematize prospective evaluation and encourage reproducibility, investigators should register analysis plans and adhere to emerging guidelines (Rivera et al., 2020; Vasey et al., 2022; Liu et al., 2020; Collins et al., 2021; Norgeot et al., 2020) for clinical AI trials. Finally, evaluation should be continuous rather than one-time. Patient populations,

care workflows, and the deployed models themselves all drift after rollout (Finlayson et al., 2021; Guo et al., 2021), so utility established at a single point can erode: a failure mode that static benchmarks, by construction, cannot surface. We therefore advocate building ongoing monitoring and evaluation directly into the deployment lifecycle (Shah et al., 2026). A discussion of designing such studies is outside the scope of this work, and we refer readers to dedicated works on this topic (Han et al., 2024; Goh et al., 2024; Omar et al., 2024; Sendak et al., 2020; Nori et al., 2023; Kim et al., 2024).

**Realign community incentives for evidence-based evaluations.** Finally, institutions and policymakers must create incentives that reward context-aware evaluation. Regulators should resist relying solely on benchmark evidence when approving systems for clinical use. Journals and conferences should require adherence to AI-specific reporting guidelines (Rivera et al., 2020; Liu et al., 2020; Vasey et al., 2022; Collins et al., 2021; Norgeot et al., 2020), and should reward real-world studies, even when results are mixed or negative. Funders should require prospective evaluation plans, public code availability and reporting of algorithm versions, and support longitudinal, interdisciplinary collaborations over short-term benchmark gains. Without systemic change, researchers will continue to optimize what is easy to measure rather than what matters.

## 5. Alternative Views

Despite their limitations, benchmarks are still important in specific aspects of the AI model development ecosystem. In this section, we discuss the use cases for which benchmarks are justified, and clarify the narrow conclusions that benchmark results can responsibly support.

**Accuracy on the Line and Scaling Laws.** Some have pointed to empirical patterns such as "accuracy-on-the-line" (Miller et al., 2021) or scaling laws (Kaplan et al., 2020; Hoffmann et al., 2022) as evidence that benchmark accuracy is predictive of real-world utility. These patterns suggest that, under certain conditions, improvements on a benchmark generalize linearly to improvements on downstream or out-of-distribution tasks. However, these findings often rely on narrow experimental setups, e.g., similar tasks or domains or synthetic shifts, and may not generalize to more complex or diverse real-world settings (Yang et al., 2024b). In high-stakes domains like healthcare, where deployment requires robustness across varied populations, workflows, and edge cases, these correlations may break down. A model's accuracy on static held-out data does not guarantee performance in clinical workflows, where uncertainty, ambiguity, and human-AI interaction play key roles. Thus, while scaling trends and benchmark predictiveness offer useful

insights during model development, they are not a substitute for context-sensitive evaluation.

**High benchmark accuracy is a prerequisite for trust and regulatory approval.** Regulatory agencies commonly call for models to be quantitatively evaluated on large, general benchmarks (Karargyris et al., 2023; Kann et al., 2021; Vokinger et al., 2021; Wu et al., 2021), and the use of general benchmarks facilitates the trust and adoption of these systems (Sourlos et al., 2024). We acknowledge that regulators and practitioners understandably look for standardized, quantitative indicators of generalizability, but emphasize the *limits* of the utility of general benchmarks. These benchmarks often use only a fraction of the data modalities available in clinical settings, such as focusing on text or images, and do not measure or encourage the use of information, such as what can be gathered through direct observation or interaction with the patient. Benchmarks also typically focus on a single metric at a time, failing to account for the wide set of axes of interest, such as fairness, safety, clarity, succinctness, and explainability. For these reasons, we argue that while benchmarks should be an aspect of evaluation, the trust placed in a system based on benchmark performance should be appropriately tempered. More robust evaluations, such as prospective studies, domain-specific safety testing, and ongoing monitoring, must also be considered.

**Benchmarks and metrics are crucial for reproducibility and scientific progress.** General benchmarks scoring protocols let different teams compare models easily and fairly, and facilitate rapid iteration (Eriksson et al., 2025). In this view, moving away from general benchmarks could lead to more disorganized research, increased difficulty in recreating and iterating on prior work, and a slower rate of progress. Without a few standard, general benchmarks, the field could lack a "common language". While we acknowledge that benchmarks can focus research efforts, they can also be a double-edged sword. If benchmarks are not aligned with clinical utility, the field can be steered towards solutions that are poorly optimized for the tasks that could provide the most clinical value and, in the worst case, towards narrow "gaming" of benchmarks.

The evaluations we call for may be harder to reproduce than standard benchmark results. Prospective and real-world studies often require institutional access, data-sharing agreements, preregistration, and deployment-specific infrastructure, which can make direct replication by reviewers or other researchers difficult. This does not make such evaluations less necessary, but it does require different norms for assessing rigor. Clinical AI evaluation may need to rely more heavily on preregistered protocols, transparent reporting, independent review of study designs, statistical standards, and replication across sites over time, rather than on the easy

re-execution common in benchmark-based ML research.

**Rather than reject benchmarks, we should make them more clinically relevant.** Even if current general benchmarks are not aligned with clinical utility, one could argue that we should simply improve them by making them more representative of the tasks and modalities important for clinical utility (Arora et al., 2025; Karargyris et al., 2023; Bedi et al., 2025a; Liu et al., 2024). We recognize the value of collecting more representative benchmarks with a clearer focus on clinical utility. However, we emphasize that no benchmark can fully reproduce a clinical context as it cannot fully capture user interactions, care workflows, or changing patient populations. Further, a model's utility depends on factors beyond data distribution, such as how physicians incorporate the model into their decision-making. Thus, while improved benchmarks reduce some risks, they do not eliminate the need for additional validation, such as more narrow domain-specific tests, prospective studies, and ongoing monitoring.

## 6. Conclusion

Benchmark performance is neither necessary nor sufficient for clinical utility. As the clinical AI community increasingly evaluates large language models and foundation models, we must avoid conflating technical performance on static datasets with the ability to improve patient care. Over-reliance on benchmarks risks distorting research incentives and delaying meaningful progress. Moving forward, we advocate for using benchmarks as an initial screening mechanism, useful for narrowing the candidate pool, but insufficient for deployment decisions. Claims of clinical utility must be supported by context-rich evaluations, such as randomized controlled trials and real-world implementation studies. By realigning evaluation standards with the needs of patients, clinicians, and health systems, we can ensure that clinical AI development remains safe, evidence-based, and socially responsible.

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
