# OpenReview forum: "Position: Benchmarks Cannot Establish Deployment Readiness of Clinical AI"
_ICML.cc/2026/Position_Paper_Track — ICML 2026 Position Paper Track regular_

### Official Review · Reviewer_3sHT · 2026-03-08

**Significance:** 2
**Argument Clarity:** 3
**Rating:** 3
**Confidence:** 4

**Questions:**

1. The paper suffers from an unbalanced target. The authors spend the vast majority of the disccusion attacking the structural flaws of static multiple-choice tests like MedQA. The modern, interactive, and multi-modal benchmarks are only simply discussed in Section 4. The authors state that even these improved benchmarks cannot fully reproduce clinical context, care workflows, or user interactions. By dismissing the field's most advanced evaluation suites in just a few sentences, the central argument disconnected from current research realities.
2. The manuscript demands extensive prospective studies. However, it neglects to explain how resource constrained computer scientists can safely screen algorithms prior to launching costly clinical trials.
3. The calls to action demand systemic shifts from funders and regulators. They fail to provide concrete or immediate methodologies for researchers building models today.
4. The definition of clinical utility is presented as a multi objective target. This concept remains too loosely defined to serve as a standardized replacement for the exact numeric metrics the authors criticize.

**Alternative Views Section:**

Yes

**Compliance With Llm Reviewing Policy A Conservative:**

Affirmed.

**Discussion Potential:**

2

**Final Justification:**

I'll keep my score.

**Paper Summary:**

This paper claims that general medical benchmarks fail to evaluate the true deployment readiness of AI systems in healthcare settings. The authors propose restricting benchmark usage to preliminary model filtering while mandating prospective trials for authentic utility claims.

**Position:**

Yes

**Position In Title:**

Yes

**Related Work:**

1

**Strengths And Weaknesses:**

The paper discusses the mismatch between static multiple choice questions and actual medical decision making.

**Support:**

2

---

> ### Author Rebuttal · Authors · 2026-03-31
>
> Thank you for the thoughtful and constructive feedback!
>
> > ...static multiple-choice tests like MedQA. ...modern, interactive, and multi-modal benchmarks... By dismissing the field's most advanced evaluation suites in just a few sentences, the central argument disconnected from current research realities.
>
> Thanks for raising this. We will engage more explicitly with modern, interactive, and multimodal benchmarks in the revision, though we note the current balance reflects their prevalence in the clinical AI literature [4]. Importantly, our core criticisms are not specific to static multiple-choice tests; they apply equally to newer benchmark designs, and we will clarify this.
>
> Our position is not that benchmarks lack value, but that even the best-designed benchmark cannot capture dynamic user interactions, care workflow integration, and evolving patient populations together sufficiently to have the needed ecological validity, and therefore cannot substitute for prospective validation and the like.
>
> We also want to be precise about who benchmarks serve. Their primary audience has historically been model developers, whose needs differ from deployment readiness, and for that purpose, benchmarks are appropriate [5-7]. However, a model is only one component of a complex healthcare system, and evaluating it in isolation from workflows and human interactions is where the limitations we identify become most consequential. Notably, as an example, NeurIPS 2026 has updated its Datasets & Benchmarks track to Evaluations & Datasets, reflecting broader evaluative needs beyond just model developers.
>
> > The manuscript demands extensive prospective studies. However, it neglects to explain how resource constrained computer scientists can safely screen algorithms prior to launching costly clinical trials.
>
> This is a fair concern, and our paper addresses it; we will make this clearer. We explicitly position benchmarks (Figure 1a) as triage tools. We are not calling for all models to enter clinical trials. Rather, benchmark success should not be interpreted as evidence warranting deployment; it is a starting point for identifying models that merit more context-specific validation.
>
> Importantly, this call is directed at stakeholders (e.g., hospitals) who may rely on benchmark performance as a primary decision criterion, rather than conducting domain-specific evaluation. It is not directed at computer scientists, who appropriately use benchmarks for model development. We agree this distinction should be more prominent and will clarify it.
>
> > The calls to action demand systemic shifts from funders and regulators. They fail to provide concrete or immediate methodologies for researchers building models today.
>
> We take this point. While Section 4 provides directional recommendations, we agree the paper would benefit from more concrete near-term suggestions and will expand this section (e.g., pre-registration for clinical AI claims, journal policies rewarding null real-world results, funder requirements for deployment evaluation).
>
> > The definition of clinical utility is presented as a multi objective target. This concept remains too loosely defined to serve as a standardized replacement for the exact numeric metrics the authors criticize.
>
> We appreciate this challenge. We intentionally define clinical utility broadly (Section 1, Figure 1b) because the required multidimensional, context-sensitive formulation cannot be captured by standard benchmarking norms. Clinical utility is indeed "fuzzy," but so are many constructs in other sciences (e.g., social sciences, psychology, educational testing), which nonetheless support rigorous evaluation practices with real-world consequences [1-3]. We will clarify that this fuzziness reflects reality rather than a weakness.
>
> [1] Wallach, Hanna, et al. "Evaluating generative ai systems is a social science measurement challenge." arXiv preprint arXiv:2411.10939 (2024).
>
> [2] Zhuang, Yan, et al. "Position: AI evaluation should learn from how we test humans." Forty-second International Conference on Machine Learning Position Paper Track. 2025.
>
> [3] Alaa, Ahmed, et al. "Medical large language model benchmarks should prioritize construct validity." arXiv preprint arXiv:2503.10694 (2025).
>
> [4] Bedi, Suhana, et al. "Testing and evaluation of health care applications of large language models: a systematic review." Jama 333.4 (2025): 319-328.
>
> [5] Donoho, David. "Data science at the singularity." Harvard Data Science Review 6.1 (2024).
>
> [6] Recht, Benjamin. "The mechanics of frictionless reproducibility." Harvard Data Science Review 6.1 (2024).
>
> [7] Hardt, Moritz, and Benjamin Recht. "Patterns, predictions, and actions: A story about machine learning."
>
> We hope these clarifications address your concerns and clarify our contributions. We would greatly appreciate your consideration in updating your evaluation of our submission, and are happy to provide additional clarification if helpful.

---

> > ### Author Rebuttal · Reviewer_3sHT · 2026-04-05
> >
> > Thanks to the authors for their reply. However, I think this paper still requires substantial updates to reflect the timeliness and usefulness in the domain. I'll keep my score.

---

### Official Review · Reviewer_XiBb · 2026-03-12

**Significance:** 4
**Argument Clarity:** 3
**Rating:** 6
**Confidence:** 3

**Questions:**

See weaknesses.

**Alternative Views Section:**

Yes

**Compliance With Llm Reviewing Policy A Conservative:**

Affirmed.

**Discussion Potential:**

4

**Final Justification:**

The authors have addressed my core concerns. I think this position paper is useful considering A* venues continually receive increasing numbers of medical eval papers.

**Paper Summary:**

The authors critique the over-reliance on benchmark scores for selection and premature integration of clinical AI models in real-world settings. Instances of benchmark-dominant models which fail in actual clinical settings are extensively highlighted. The limitations of benchmarks, such as dataset flaws, inability to probe natural multi-turn investigation, and memorization are discussed. The authors advocate for the usage of benchmarks as an initial tool for model selection, which must then undergo prospective studies, rigorous real-world studies, and continuous monitoring. They also propose strategies for higher-quality benchmark design and to incentivize the pursuit of research which actually performs real-world studies.

**Position:**

Yes

**Position In Title:**

Yes

**Related Work:**

3

**Strengths And Weaknesses:**

Strengths:

- The authors convincingly describe why benchmarks are unsuitable for predicting real-world clinical performance. In particular, the authors comprehensively contrast the design/objectives of benchmarks and actual clinical tasks.

- Extensive examples of "good models" failing in actual deployment with citations.

- Rather than entirely discounting benchmarks, the authors describe how benchmarks can be used as a simple and initial component for further exploration of model adoption.

Weaknesses:

- The authors could devote a bit more discussion to the current landscape and evolution of clinical models. The models discussed are already slightly dated. For example, ChatGPT Health or any other modern frontier model.

- The authors skirt over the reproducibility problem. Namely, in venues such as ICML, ICLR, NeurIPS, etc., we expect that the reader is capable of reproducing all results (emphasis on "expect"), whereas a real-world study/RCT for a new clinical model cannot easily be verified. Of course, medical journals already function under the expectation of non-immediate reproducibility and clearly advance medicine anyways, so the authors should make suggestions on how AI/ML venues can accommodate papers which contain real-world results that cannot easily be verified by reviewers.

**Support:**

4

---

> ### Author Rebuttal · Authors · 2026-03-31
>
> Thank you for the thoughtful and constructive feedback! We are glad you found our work to be “convincing and comprehensive.”
>
> > The authors could devote a bit more discussion to the current landscape and evolution of clinical models. The models discussed are already slightly dated. For example, ChatGPT Health or any other modern frontier model.
>
> Thank you for the helpful feedback. We acknowledge that our discussion of deployed models skews toward general-purpose models and older health-specific systems, as these have the most well-documented real-world failure data. We will add discussion of more recent health-specific frontier model deployments and their clinical evaluations, many of which have been published since our submission, including ChatGPT Health in clinical workflows and recent clinical trial results. We agree that this would strengthen the paper's relevance and timeliness significantly.
>
> > The authors skirt over the reproducibility problem. Namely, in venues such as ICML, ICLR, NeurIPS, etc., we expect that the reader is capable of reproducing all results (emphasis on "expect"), whereas a real-world study/RCT for a new clinical model cannot easily be verified. Of course, medical journals already function under the expectation of non-immediate reproducibility and clearly advance medicine anyways, so the authors should make suggestions on how AI/ML venues can accommodate papers which contain real-world results that cannot easily be verified by reviewers.
>
> This is an interesting tension you raise; thank you. You are right that conferences like ICML have norms around reproducibility, particularly the ease of it, which has many attractive properties [1-3]. However, the more involved evaluations we call for create institutional friction that limits reproducibility and replication by reviewers and others. We will add a dedicated discussion of this point, drawing on how other fields have addressed this challenge via pre-registration, data-sharing agreements, independent review of methodology rather than re-execution of results, and editorial standards around statistical reporting. Venues like ICML could adopt analogous practices, for instance, a more formal assessment of study design rigor rather than requiring the usual re-execution, or establishing a category of "real-world evaluation" papers with adapted review criteria. Ultimately, the ease of replication that characterizes typical AI/ML work is unlikely to be maintained with the types of evaluations our work calls for, and we see developing community norms around this as an important open challenge.
>
> [1] Donoho, David. "Data science at the singularity." Harvard Data Science Review 6.1 (2024).
>
> [2] Recht, Benjamin. "The mechanics of frictionless reproducibility." Harvard Data Science Review 6.1 (2024).
>
> [3] Hardt, Moritz, and Benjamin Recht. "Patterns, predictions, and actions: A story about machine learning."
>
> We hope that our clarifications have addressed your concerns and helped clarify our contributions. We are, of course, happy to provide any additional clarification if helpful.

---

> > ### Author Rebuttal · Reviewer_XiBb · 2026-04-03
> >
> > Thanks for the response. I am satisfied and will maintain my positive score.

---

### Official Review · Reviewer_23fs · 2026-03-13

**Significance:** 2
**Argument Clarity:** 3
**Rating:** 3
**Confidence:** 3

**Questions:**

See weakness

**Alternative Views Section:**

Yes

**Compliance With Llm Reviewing Policy A Conservative:**

Affirmed.

**Discussion Potential:**

2

**Final Justification:**

Part of my concerns are resolved, but I still believe this paper's position is not novel enough, as link LLM with practical clinical is always a hot topic in Meidcal AI.

**Paper Summary:**

The paper argues that benchmark performance does not measure deployment readiness in clinical AI, and proposes that benchmarks should only be used as an initial filter for candidate models, while claims about clinical utility should rely on prospective real-world evaluation such as clinical trials and deployment studies.

**Position:**

Yes

**Position In Title:**

Yes

**Related Work:**

3

**Strengths And Weaknesses:**

Strength

1. The topic is important, where clinical AI deployment is high-stakes and evaluation methodology is a central issue in ML.

2. The paper is generally well organized and clearly written.


Weakness

1.  This position fits the scope of the ICML position track. However, the position itself is not particularly novel, as similar arguments have already been widely discussed in previous evaluation literature. Such as clarity [1,2] or other real-world requirements [3].

2. Most ML and healthcare researchers already agree that benchmark performance ≠ deployment readiness and  real-world validation is required [1,2,3]. Especially, a previous study has arouse the attention for the readiness [4].


[1] MedReason: Eliciting Factual Medical Reasoning Steps in LLMs via Knowledge Graphs

[2] BUILDING A HUMAN-VERIFIED CLINICAL REASONING DATASET VIA A HUMAN–LLM HYBRID PIPELINE FOR TRUSTWORTHY MEDICAL AI

[3] GAPS: A Clinically Grounded, Automated Benchmark for Evaluating AI Clinicians

[4] The Illusion of Readiness in Health AI

**Support:**

2

---

> ### Author Rebuttal · Authors · 2026-03-31
>
> Thank you for the thoughtful and constructive feedback! We are glad you found our work to address an “important topic” and to be “well organized and clearly written.” Below, we address what we believe are misreadings of our positions to clarify our contribution.
>
> > This position fits the scope of the ICML position track. However, the position itself is not particularly novel, as similar arguments have already been widely discussed in previous evaluation literature. Such as clarity [1,2] or other real-world requirements [3].
>
>
> Thank you for the references! We note that the references [1-3] operate within the existing benchmark paradigm, while our paper challenges it. Thus, we believe that these papers don't preempt our position, but actually demonstrates the need for it. For example, [1] and [2] propose new benchmarks to measure progress, but don't question whether benchmark performance translates to clinical utility and real-world decision making for a fixed model. GAPS [3] does propose a richer benchmarking framework, and we discuss similar initiatives in Section 4 of our paper. However, it still addresses quality issues within the benchmarking paradigm, whereas we argue that this paradigm itself is insufficient for deployment.
>
>
>
> > Most ML and healthcare researchers already agree that benchmark performance ≠ deployment readiness and real-world validation is required [1,2,3]. Especially, a previous study has arouse the attention for the readiness [4].
>
>
> We respectfully disagree that this is the case, and though prior studies [4] have studied deployment readiness of clinical LLMs, we do not believe that this position is common in the ICML/machine learning community. If the community agreed that benchmarks do not measure deployment readiness, we would expect to see this reflected in research practices and publication norms. However, benchmark performance is virtually the only metric by which clinical AI methods are evaluated.
>
> More importantly, the over-reliance of benchmarks for deployment decisions also extends to the regulatory and policy landscape. For example, the EU AI Act [5] mandates "conformity assessment" (Article 43) for high-risk AI, but does not require demonstration of real-world clinical utility through deployment studies. In fact, in many scenarios such as clinical decision support, an internal self-assessment is sufficient, without independent third party review. Prospective validation is also not specified in California's recently enacted healthcare AI laws (SB 1120). Finally, the FDA's 2026 Clinical Decision Support [6] guidance exempts certain AI tools from device regulation altogether, as long as healthcare professional can independently review the recommendation. Indeed, clinician-facing LLMs are already being deployed in large academic medical centers [7], and it is possible for such models to be deployed without any prospective validation.
>
>
> [5] https://eur-lex.europa.eu/eli/reg/2024/1689/oj/eng
>
> [6] https://www.fda.gov/regulatory-information/search-fda-guidance-documents/clinical-decision-support-software
>
> [7] https://arxiv.org/abs/2602.00074
>
> We hope that our clarifications have addressed your concerns and helped clarify our contributions. We would greatly appreciate your consideration in updating your evaluation of our submission. We are, of course, happy to provide any additional clarification if helpful.

---

> > ### Author Rebuttal · Reviewer_23fs · 2026-04-04
> >
> > I'm not convinced by **position is not common in the ICML/machine learning community**, as there is a huge area in studying trustworthy medical AI. In addition, [1] and [2]  propose methods for generating good data to train a model, and they involve medical experts in the design, thus indeed consider clinical utility and real-world decision making.

---

### Official Review · Reviewer_2jpU · 2026-03-16

**Significance:** 2
**Argument Clarity:** 2
**Rating:** 3
**Confidence:** 4

**Questions:**

As discussed in the weakness points, this paper is relatively shallow, not sufficientl scientific contributions to be accepted by ICML.

Weaknesses

1. This work calls for prospective studies. However, it lacks specific design guidelines. This gap may hinder implementation by clinical teams.
2. This work effectively highlights benchmarking limitations. Yet it offers limited discussion on improving benchmarks. How can benchmarks better approximate clinical reality? This question deserves more attention.
3. This work mentions the need to adjust community incentives. However, it lacks actionable policy recommendations. More specific guidance would strengthen this point.

**Alternative Views Section:**

Yes

**Compliance With Llm Reviewing Policy A Conservative:**

Affirmed.

**Discussion Potential:**

2

**Ethics Review Area:**

["Other Expertise"]

**Paper Summary:**

This position paper presents a key argument: while large language models (LLMs) excel on benchmark tasks like medical question-answering, these benchmarks do not accurately measure clinical deployment readiness. The authors observe a recurring pattern: many healthcare AI systems that perform well on retrospective accuracy tests fail in real-world deployment, while some systems with modest benchmark performance deliver genuine clinical benefits. The paper provides a detailed analysis of why benchmarks misalign with clinical tasks. Firstly, clinical decision-making is dynamic and iterative, yet most benchmarks are static. Secondly, real-world care integrates multi-modal information (e.g., notes, labs, imaging, and patient context), while benchmarks often rely on simplified, single-modality inputs. Finally, evaluation metrics in benchmarks are frequently oversimplified, focusing on accuracy rather than clinical impact. The authors conclude that benchmarks should serve only as preliminary screening tools. True clinical utility must be validated through prospective, real-world studies.

**Position:**

Yes

**Position In Title:**

Yes

**Related Work:**

3

**Strengths And Weaknesses:**

Strengths

1. The paper accurately identifies the overreliance on benchmarking in clinical AI evaluation. This insight is crucial for preventing resource waste and, more importantly, for safeguarding patient safety.
2. The authors ground their argument with well-chosen examples. Failures like IBM Watson for Oncology and the Epic sepsis model illustrate the risks of benchmark-driven development. Successes like Ambient AI Scribe show that clinical value can emerge even without top-tier benchmark scores. These cases powerfully support their claim.
3. The paper critiques the current state of clinical AI evaluation. It also proposes a concrete roadmap. Key suggestions include three directions: using benchmarks as triage tools; designing benchmarks around decision-making processes; and conducting prospective utility testing..

Weaknesses

1. This work calls for prospective studies. However, it lacks specific design guidelines. This gap may hinder implementation by clinical teams.
2. This work effectively highlights benchmarking limitations. Yet it offers limited discussion on improving benchmarks. How can benchmarks better approximate clinical reality? This question deserves more attention.
3. This work mentions the need to adjust community incentives. However, it lacks actionable policy recommendations. More specific guidance would strengthen this point.

**Support:**

2

---

> ### Author Rebuttal · Authors · 2026-03-31
>
> Thank you for the thoughtful and constructive feedback! We are glad you found our work to provide a “crucial insight for safeguarding patient safety,” “ground our argument in well-chosen examples,” and “proposes a concreate roadmap” to address identified limitations.
>
> > This work calls for prospective studies. However, it lacks specific design guidelines. This gap may hinder implementation by clinical teams.
>
> We thank the reviewer for this suggestion! However, as we have noted on L372-375, a discussion on designing prospective studies is out of the scope of this work, as there are many prior works in this area. Our contribution is to argue when and why the field should move towards such evaluations, not to duplicate what these existing works already cover on the methodological details of these evaluations.
>
>
>
>
>
> > This work effectively highlights benchmarking limitations. Yet it offers limited discussion on improving benchmarks. How can benchmarks better approximate clinical reality? This question deserves more attention.
>
>
> First, we emphasize that we *do* address several directions for benchmark improvement in Section 4. We discuss more-realistic EHR-based QA settings with "noisy, multi-format data [which] requires reasoning across encounters" (L300-302), simulation frameworks with interactive patient models (L304), the conversion of static QA into interactive settings (L310), and broader leaderboards such as MedHELM and HealthBench (L311-315). We also explicitly recommend designing benchmarks around clinical decisions rather than factual recall (L332-347).
>
> However, the core position of our paper is that benchmark improvement alone is inherently insufficient, as "no benchmark can fully reproduce clinical context as it cannot fully capture user interactions, care workflows, or changing patient population" (L414-417). Thus, overemphasizing benchmark improvement in our paper is antithetical to the benchmark-centrism we critique.
>
>
>
> > This work mentions the need to adjust community incentives. However, it lacks actionable policy recommendations. More specific guidance would strengthen this point.
>
>
> We respectfully disagree that the paper lacks actionable policy recommendations. We have listed several concrete directions in Section 4, including (1) "Regulators should resist benchmark-only evidence when approving systems for clinical use." (L381), (2) "Journals and conferences should reward real-world studies, even when results are mixed or negative." (L382), and (3) "Funders should support longitudinal, interdisciplinary collaborations over short-term benchmark gains." (L384). These directions target major stakeholders: regulatory bodies, publication venues, and funding agencies. We are happy to expand on this discussion in the revision with more examples, but given that his is a position paper at a machine learning venue, we feel like the level of policy discussion is well-calibrated to the audience.
>
>
>
>
> > As discussed in the weakness points, this paper is relatively shallow, not sufficientl scientific contributions to be accepted by ICML.
>
>
> We note that the ICML Call for Position Papers states that position papers should be "judged primarily on whether they present a compelling position that warrants greater exposure within the machine learning community", not whether they present "original research and novel results". We believe that our paper meets this standard, and we are glad the reviewer has acknowledged that our paper "accurately identifies the overreliance on benchmarking in clinical AI evaluation" and that our paper "proposes a concrete roadmap".
>
> We hope that our clarifications have addressed your concerns and helped clarify our contributions. We would greatly appreciate your consideration in updating your evaluation of our submission. We are, of course, happy to provide any additional clarification if helpful.

---

> > ### Author Rebuttal · Reviewer_2jpU · 2026-04-06
> >
> > I think the contributions of this work is more fundamentally limited thus I will keep the score.

---

### Decision · Program_Chairs · 2026-04-30

**Decision:**

Accept (regular)

**Comment:**

** Overall: ** This paper clearly tackles a specific issue in clinical applications for models, and makes a compelling argument against relying solely on static benchmarks. The primary weakness of the paper is that it devotes significant effort to the least objectionable part of the argument, that static benchmarks are insufficient. The portion that has the most discussion potential (what policies help change this, what kinds of studies would be sufficient, to what degree more recent interactive benchmarks already address the issues) receives much less attention in the paper, and thus lowers the discussion potential. However, the authors raise many very valuable points and have laid out a sufficiently clear roadmap that there is still high discussion potential for this paper, and it has many concrete and impactful implications.

** Primary strengths: **
- The paper clearly identifies issues that come with over-reliance on benchmarks, particularly for real world settings and deployment decisions (2jpU, XiBb)
- The risks discussed are concrete and focus on actual examples of meaningful failures (2jpU), and these examples focus on cases where “good” models failed (XiBb).
- The topic is of clear importance (23fs) and identified a genuine mismatch between benchmarks possible utility and what they are ultimately used for (3sHT)

** Primary weaknesses: **
- The paper focuses more on known problems than on a concrete path forward. Though the paper calls for prospective studies and policy changes, details on what this actually means are weak (2jpU). The authors argue that this is out of scope and they don’t want to replicate extensive past work in laying out the methodology for prospective studies (I agree with the authors’ choice here), but this seems to misunderstand the reviewer’s point that discussion pontential is limited without a more concrete proposal. Reviewer 3sHT brings up a similar point, noting that the focus of the paper is unbalanced towards identifying the issues with bencharks (and issues with benchmarking have been widely discussed), but further raising the issue that much of this critique is focused on static benchmarks even as interactive benhcmarking has become much more prevalent.
- Lack of novelty of the core argument that benchmarks are insufficient (23fs). Though as the authors argue novel argumentation or resutls are not a direct requirement for position papers, the issue of evaluation validity in real world settings *has* received broad attention, particularly in the medical AI community, and it’s unclear what the additional discussion on this particular paper would be within the ICML community. The authors do raise the valid point that research norms often do not yet address the insufficiencies in benchmarking, but that does not mean that the awareness of the issue is not already widespread, making it less clear what contribution further discussion has.
- The proposed solutions have a tension with the expectation of reproducibility (XiBb). The authors agree to add discussion on this point centered around how ICML can adapt its reviewing guidelines in ways that mirror other fields.